# Proximal renal tubular function in HIV-infected children on tenofovir disoproxil fumarate for treatment of HIV infection at two tertiary hospitals in Harare, Zimbabwe

Runyararo Mashingaidze-Mano [1]*, Mutsawashe F. Bwakura-Dangarembizi[1], Charles C. Maponga[2,3,4], Gene D. Morse[3], Tsitsi G. Monera-Penduka [2], Takudzwa J. Mtisi [5], Tinashe Mudzviti [2,4,6], Hilda A. Mujuru [1]

1 Department of Paediatrics, University of Zimbabwe College of Health Sciences, Harare, Zimbabwe, 2 School of Pharmacy, University of Zimbabwe College of Health Sciences, Harare, Zimbabwe, 3 Center for Integrated Global Biomedical Sciences, University at Buffalo, New York, United States of America, 4 International Pharmacotherapy Education and Research Initiative, Harare, Zimbabwe, 5 Department of Clinical Pharmacology, University of Zimbabwe College of Health Sciences, Harare, Zimbabwe, 6 Newlands Clinic, Highlands, Harare, Zimbabwe

* runyamano@yahoo.com, drmarshy@yahoo.com

**Data Availability Statement:** All relevant data are within the paper and its Supporting Information files.

## Abstract

### Background

Renal abnormalities in HIV infected children may be due to the HIV infection or treatment among other factors. Tenofovir disoproxil fumarate (TDF) is associated with proximal renal tubular dysfunction, proteinuria and decrease in glomerular function. Studies in developed countries have shown variable prevalence of proximal renal tubular dysfunction in children on TDF. There are no known studies in developing countries, including Zimbabwe, documenting the proximal tubular function in HIV infected children on TDF. The aim of this study was to assess renal and proximal renal tubular function in HIV infected children receiving TDF and determine factors associated with proximal tubular dysfunction.

### Methods

A descriptive cross-sectional study was conducted in HIV infected patients below 18 years of age attending outpatient clinics at two tertiary hospitals in Harare, who received a TDF-containing antiretroviral regimen for at least six months. Dipstick protein and glucose, serum and urine phosphate and creatinine levels were measured. Fractional excretion of phosphate was calculated. Estimated glomerular filtration rate (eGFR) was calculated using the Schwartz formula. Tubular dysfunction was defined by at least two of the following characteristics: normoglycaemic glycosuria, hypophosphatemia and fractional excretion of phosphate > 18%.

**Funding:** Runyararo Mashingaidze-Mano received funding through the Fogarty International Center of the National Institutes of Health (grant numbers D43TW010313 -01 and D43TW007991), for this research as part of the Fogarty International Centre under University at Buffalo [UB] and the University of Zimbabwe (UZ) HIV Research Training Program (HRTP). The funders had no role in the study design, data collection and analysis, decision to publish or preparation of the manuscript.

**Competing interests:** The authors declare that no competing interests exist.

**Abbreviations:** ABCC2, ATP binding cassette subfamily C member 2; B2M, β-2 microglobulin; DAIDS, Division of acquired immunodeficiency syndrome; FEp, Fractional excretion of phosphate; IQR, Interquartile range; KDOQI, Clinical Practice Guidelines for Chronic Kidney Disease: Evaluation, Classification and Stratification; MRCZ, Medical Research Council of Zimbabwe; NAG, N-acetyl-β-d-glucosaminidase; PI, Protease Inhibitor; RBP, Retinol binding Protein; TAF, Tenofovir alafenamide; TDF, Tenofovir disoproxil fumarate; TPR, Tubular phosphate reabsorption; uARP, Urine albumin/total protein ratio.

## Findings

One hundred and ninety-eight children below 18 years of age were recruited over a period of six months. The prevalence of tubular dysfunction was 0.5%. Normoglycaemic glycosuria occurred in 1 (0.5%), fractional excretion of phosphate >18% in 4 (2%), and hypophosphatemia in 22 [11.1%] patients. Severe stunting was associated with increased risk of hypophosphatemia (OR 9.31 CI (1.18, 80.68) p = 0.03). Reduction in estimated glomerular filtration rate (eGFR) < 90ml/min/1.73m$^2$ and proteinuria was evident in 35.9% and 32.8% of children, respectively. Concurrent TDF and HIV-1 protease inhibitor-based regimen was the only independent factor associated with reduction in GFR (OR 4.43 CI (1.32; 4.89) p = 0.016).

## Conclusion

Tubular dysfunction was uncommon in Zimbabwean children on a TDF-based ART regimen. Hypophosphatemia, proteinuria and reduction in eGFR were common. Reassessing renal function using more sensitive biomarkers is needed to examine the long-term tolerance of TDF.

## Introduction

Tenofovir disoproxil fumarate (TDF), a nucleotide reverse transcriptase inhibitor approved by the World Health Organization (WHO) for the management of HIV infection in children is associated with nephrotoxicity [1]. Toxicity can manifest as proximal tubular or glomerular dysfunction, acute kidney injury, chronic kidney disease and end stage renal disease [2]. Mitochondrial damage to the proximal tubule leads to impaired reabsorption of low molecular weight proteins and other solutes resulting in urinary wasting. TDF-associated tubular dysfunction may affect the serum bicarbonate and glucose concentration and in severe cases lead to Fanconi syndrome, a generalized proximal tubular defect with hypokalemic metabolic acidosis associated with vitamin D resistant metabolic bone disease [3, 4].

Advanced HIV infection and HIV-1 protease inhibitor (PI) containing therapy, are among factors associated with TDF renal impairment [5]. Biomarkers used to assess proximal renal tubular function include; urinary low molecular proteins serum phosphates, phosphaturia, normoglyacemic glycosuria, as well as urinary and serum electrolytes [6]. In Zimbabwe TDF is recommended as part of the first line regimen for adolescents and children weighing at least 25kg [7]. Ideally, children on TDF-based regimen should be monitored for creatinine clearance [8] but in most developing countries such as Zimbabwe, that recommendation is constrained by limited resources. Studies describing renal tubular function in children on TDF are therefore limited to those mostly assessing glomerular function in ART-naïve cases [9, 10]. This study aimed to assess the prevalence and factors associated with proximal renal tubular dysfunction in children receiving TDF. It was assumed that about 80% of the children had vertical transmission of HIV based on a study by Ferrand *et al* [11].

## Methods

### Study design and participants

A cross-sectional study was conducted between May and November 2016 at two Paediatric HIV Clinics in Harare, Zimbabwe. The Institutional Review Boards of: Harare Central

Hospital Ethics Committee (Ref: HCHEC 140916/62), the Joint Research Ethica Committee For The University of Zimbabwe College of Health Sciences and Parirenyatwa Group of Hospitals (JREC), (Ref: JREC66/16) and the Medical Research Council of Zimbabwe (MRCZ), (Ref: MRCZ/B/1053) approved the study. Written informed consent was acquired from caregivers and assent obtained from children aged six years and older. Children less than 18 years who weighed at least 25 kg and were on a TDF based regimen for at least six months were recruited during routine clinic visits, conducted on Mondays to Thursdays inclusive. A convenience sampling method was used. Patients were recruited in the outpatients clinic as they came either for doctor's review, medicine pickup or blood collection. Those patients with pre-existing diabetes or hypertension were excluded from the study. The Dobson formula below was used to calculate sample size:

$$n = \left(\frac{Z_{\alpha/2}}{d}\right)^2 p(1-p)$$

where $Z_{\alpha/2}$ is the standard normal value corresponding to the desired level of confidence (95%)

  **d** is the maximum allowable error, (0.05) or 5% (width of the confidence intervals)

  **p** is the estimated prevalence of renal tubular dysfunction in children on TDF based regimen The calculated sample size was 196 participants based on a study in Ghana which found a prevalence **(p)** of proximal renal tubular dysfunction of 15% [12]. The precision for the sample size was 5%.

## Clinical assessment

Demographic data, baseline CD4 count and World Health Organization (WHO) clinical staging were captured from participants and medical records respectively prior to highly active antiretroviral therapy (HAART) initiation. Blood pressure was measured in a sitting position using an appropriate size cuff (Dynamap V100™ manufactured by Menhold South Africa). Weight and height measurements were taken by a physician or trained clinic nurse. Weight was measured using a calibrated scale (Seca, model 8811021659) estimated to the nearest 0.1kg. Height was measured using a standardized wall mounted stadiometer with inbuilt millimeter ruler and estimated to the nearest 0.1cm. Body mass index calculation was done using the WHO AnthroPlus calculator (2009) [13]. The WHO growth charts (2007) were used to determine nutritional status of the study participants [14–16].

## Laboratory assessment

Spot urine samples were collected during the clinic visits. The urine was immediately checked for presence of glucose and protein using the 10 parameter reagent strips (Uricheck M10, Omnipharm). Patients with positive urine glucose (>+1), had a capillary blood glucometer performed immediately (Glucoplus™). Qualitative proteinuria was reported on dipstick as negative, 1+(30mg/dL), 2+ (100mg/dL), 3+(300mg/dL) or 4+(1000mg/dL) [17]. Urine protein/creatinine ratio (mg/dL:mg/dL) was further performed on all urine samples. Proteinuria was defined as; normal range proteinuria if urine protein/creatinine <0.2g, further classified into intermediate proteinuria (0.2g - 3.0g) and nephrotic range proteinuria (>3.5g) [18]. The rest of the urine samples including glucose or protein negative dipstick were kept in a carrier cooler bag lined with ice packs between +2°C and +8°C then transported to the University of Zimbabwe, Department of Chemical Pathology laboratory within four hours of collection, for assessment of urine phosphate, creatinine and protein. Blood samples for measurement of serum creatinine and phosphate were collected and transported to the laboratory in carrier bags lined with ice packs maintaining temperature between +2°C and +8°C. Specimens were

processed within same day of collection. Urine and serum phosphorus concentration was measured using the Phosphomolybdate method [19]. The pyrogallol red method was used to measure urine protein concentration [20]. Serum and urine creatinine were measured using the modified Jaffe method [21, 22]. Estimated glomerular filtration rate (eGFR) was calculated using the Schwartz formula [23, 24]. A Mindray BS200E Chemistry analyser (Mindray, Shenzhen, China) was used to determine the concentrations of creatinine, phosphate and protein for both urine and serum samples. Calibration of the machine was done every fortnight for creatinine and monthly for phosphate and protein. Proximal tubular function was assessed using normoglyacemic glycosuria, fractional excretion of phosphate (FEp) >18% and hypophosphatemia. The FEp was calculated using a standard formula [25]. Hypophosphatemia was graded according to the Division of Acquired Immunodeficiency Syndrome (DAIDS) toxicity grading [26, 27]. Glomerular dysfunction as indicated by a low eGFR was classified as mild (60–89 ml/min/1.73 $m^2$), moderate (30–60 ml/min/1.73 $m^2$), and severe impairment (15–30 ml/min/1.73 $m^2$) [8, 28]. The full laboratory protocol is found at protocols.io: https://protocols.io/view/pone-d-19-35784-proximal-renal-tubular-function-in-bfzxjp7n [dx.doi.org/10.17504/protocols.io.bfzxjp7n]

**Table 1. Demographic and clinical characteristics of HIV infected children, on TDF for at least 6 months, < 18 years old, (n = 198).**

| Variable | Frequency n = 198 (%) |
|---|---|
| **Gender** | |
| Female | 89 (44.9) |
| Male | 109 (55.1) |
| Age | |
| Median age [years] | 15(IQR 13–16) |
| CD4 count ul/l | |
| ≤ **200** | 63(31.8) |
| **201–500** | 25 (12.6) |
| **501–800** | 24 (12.1) |
| ≥ **800** | 11 (5.6) |
| WHO Clinical Stage | |
| **Stage 1** | 41 (20.7) |
| **Stage 2** | 32 (16.1) |
| **Stage 3** | 61 (30.8) |
| **Stage 4** | 13 (6.6) |
| Nutritional Status | |
| *BMI for age [kg/$m^2$] | |
| <-2SD | 14 (7.0) |
| <-3SD | 11 (5.6) |
| ≥1SD | 10 (5.1) |
| ≥2SD | 3 (1.5) |
| >-2SD | 160 (80.8) |
| Stunting *[HFA] | |
| <-2SD* | 26 (13.1) |
| <-3SD | 18 (9.1) |
| >-2SD | 154 (77.8) |

*BMI = Body mass index, HFA = Height for age, SD = Standard deviation.

## Data analysis

Data were analyzed using Epi Info version 7. Descriptive statistics with median and interquartile range (IQR) for continuous, non-nominal variables and percentages for categorical variables were generated. Univariate and multivariate logistic regression analyses were performed to identify factors associated with proximal renal tubular dysfunction. Factors with p < 0.25 in the univariate analysis were entered into a stepwise multivariate analysis model. The results of the logistic regression analyses were expressed as odds ratios (ORs) with 95% confidence intervals(CIs). The MDB file data sets can be found at https://datadryad.org/stash/share/pRmK7SDWzk5Jnd3-usw2hKCGVe71OksxVKbxfTZv5PE and DOI (doi:10.5061/dryad.2fqz612mf).

## Results

One hundred and ninety-eight children were enrolled into the study. The demographic and (clinical characteristics are summarized in Tables 1 and 2. The median age of participants was 15 years (IQR 13–16; Range 6–17.11). The median duration on a TDF based regimen was 37 months (IQR 16–52; Range 6–120). Baseline CD4 count was documented in 123 /198 children (62.1%). Fourteen (7%) participants' BMI was classified as thin, 9 (5.6%) as severe thinness and 13 (6.6%) as overweight. Severe stunting (height for age <-3standard deviation) was recorded in 18 (9.1%).

Table 3 below shows the frequency of abnormal renal function tests. These tests include serum phosphate decrease, fractional excretion of phosphate >18%, normoglycaemic glycosuria, reduction in glomerular filtration rate and proteinuria.

## Proximal renal tubular function

FEp> 18% was detected in 4 (2%), hypophosphatemia in 22 (11.1%) and normoglycaemic glycosuria in one (0.5%) participant. Mild to moderate decrease in serum phosphate was detected in 16 (8%) of the children while 6 (3%) had severe to life threatening hypophosphatemia. Tubular dysfunction was only detected in one male patient who had FEp> 18% and hypophosphatemia. This patient was older than 14 years and had been on first line ART for 55 months. Severe stunting was associated with increased risk of hypophosphatemia in children younger than 14 years (OR 9.31 CI (1.18, 80.68) p = 0.03). There were no significant factors associated with hypophosphatemia in children older than 14 years (Tables 4 and 5 below).

**Table 2. ARV exposure of HIV infected children, on TDF for at least 6 months, < 18 years old, (n = 198).**

|  | N 198 (%) |
|---|---|
| ART experienced prior to TDF use | 107 (54) |
| ART- naïve prior to TDF use | 91 (46) |
| No. of patients First Line Therapy | 165 (83.3) |
| No. of patients on Second Line Therapy | 33 (16.7) |
| Current ART regimen |  |
| TDF/3TC/EFV | 139 (70.2) |
| TDF/3TC/NVP | 26 (13.1) |
| TDF/3TC/bPI* | 33 (16.7) |
| Median duration on TDF (months) | 37 (IQR 16–52) |
| Range (months) | 6–120 |

*bPI is either Lopinavir/ritonavir or Atazanavir/ritonavir.**

**Table 3. Proximal renal tubular function; urinary and serum biomarkers and glomerular function abnormalities of HIV infected children, on TDF for at least 6 months, < 18 years.**

| Markers of renal function | n [%] |
|---|---|
| FEp>18%* | 4 (2) |
| Normoglycemic Glycosuria | 1 (0.5) |
| Serum Phosphate Decrease | 22 (11.1) |
| *eGFR<90ml/min² | 71 (35.9) |
| Proteinuria | 65 (32.8) |

*FEp = Fractional excretion of phosphate, *eGFR = estimated glomerular filtration rate.

## Proteinuria

Proteinuria with at least 1+ of protein was detected in 31(15.7%) of study participants on dipstick urinalysis. Only one participant had glycosuria on dipstick. Of the 31 participants with positive dipstick proteinuria only 7 (3.6%) were confirmed by spot urine protein:creatinine ratio. Sixty-five of the 193 samples were positive for proteinuria increasing positive to 33.7% (65/193), despite negative dipstick urinalysis in most of the participants. Nephrotic range proteinuria was detected in only 2 (1%) of the children. A protease inhibitor (PI) containing ART regimen was significantly associated with proteinuria in both univariate (OR 3.60 CI (1.62, 7.99) p = 0.002*) and multivariate analysis (0R 3.75 CI (1.59; 8.86) p = 0.003), Fig 1. Duration on TDF was not significantly associated with proteinuria. Children with moderate and severe stunting were more likely to have proteinuria though not statistically significant (Fig 1).

## Glomerular filtration rate

Estimated GFR > 90ml/min/m² was in 122 (64.2%) of the participants. Among the children with decreased eGFR, 67 had mild, three moderate and one had severe reduction. Exposure to a protease inhibitor (PI) was associated with reduction in eGFR and this was statistically significant (OR 6.08 CI (2.56, 14.45) p<0.001), Fig 2. On multivariate analysis, exposure to PI regimen was the only independent factor associated with reduction in eGFR (p = <0.016) Patients with severe stunting, height for age (HFA) <-3 were more likely to have reduction in eGFR (OR 2.69 CI (1.00, 7.24) p = 0.051). Although age had a p-value greater than 0.25 in the univariate model it was included in the multivariate model to account for any confounding effects age may have on eGFR.

## Discussion

This study uniquely describes the prevalence of proximal renal tubular dysfunction in paediatric patients on TDF based ART regimen in a resource constrained clinical setting.

## Renal tubular function and dysfunction

There was a low prevalence of proximal tubular dysfunction found in the study population. Similar findings were reported in two studies of children treated with TDF in two developed countries, where renal tubular function remained stable throughout the study period [29, 30]. However, finding in this study were much lower than what was reported in a multicenter study in Spain where tubular dysfunction measured by reduction in tubular phosphate reabsorption (TPR) was observed in 74% and proteinuria 89% of study participants [31]. The study design, definitions and parameters of proximal tubular dysfunction used in the Spanish study may explain the differences. In a study by Chadwick et al and Labarga *et al* in Ghana, despite

**Table 4. Factors associated with hypophosphatemia in children <14 years on TDF regimen for at least 6 months.**

| VARIABLE | CATEGORY | YES | NO | OR (95%CI) | P-VALUE |
|---|---|---|---|---|---|
| Gender | Female | 5 (31.3) | 34 (46.6) | | |
| | Male | 11 (68.7) | 39 (53.4) | 0.52 [0.17, 1.70] | 0.263 |
| Current ART Regimen. | TDF/3TC/EFV | 10 (62.5) | 54 (74) | 1 | |
| | TDF/3TC/NVP | 2 (12.5) | 8 (11) | 1.35 [0.24, 7.32] | 0.728 |
| | TDF/3TC/PI | 4 (25) | 11 (51.1) | 1.93 [0.52, 7.41] | 0.451 |
| ART Regimen | 1st Line | 13 (81.2) | 63 (86.3) | | |
| | 2nd Line | 2 (18.8) | 10 (13.7) | 1.03 [0.2, 5.27] | 1 |
| Duration on current regimen in months | 6–12 | 4 (25) | 9 (12.3) | 1 | |
| | 13–60 | 12 (75) | 63 (86.3) | 2.33 [0.62, 8.82] | 0.212 |
| | ≥61 | 0 | 1 (1.4) | - | - |
| W.H.O Staging | Stage 1 | 5 (45.5) | 15 (24.2) | 1 | |
| | Stage 2 | 2 (18.2) | 15 (24.2) | 0.40 [0.06, 2.39] | 0.416 |
| | Stage 3 | 4 (36.4) | 27 (43.5) | 0.44 [0.10, 1.91] | 0.289 |
| | Stage 4 | 0 | 5 (8.1) | - | - |
| *BMI | Normal | 12 (75) | 60 (82.2) | 1 | |
| | <-2SD | 2 (12.5) | 6 (8,2) | 1.67 [0.30, 9.27] | 0.623 |
| | <-3SD | 1 (6.3) | 4 (5.5) | 1.25 [0.13, 12.19] | 1 |
| | >1SD | 0 | 2 (2.7) | - | - |
| | >2SD | 1 (6.3) | 1 (1.4) | 5.0 [0.29, 85.60] | 0.322 |
| Stunting | >-2SD | 11 (68.8) | 67 (91.8) | 1 | |
| | <-2SD | 2 (12.5) | 4 (5.5) | 3.05 [0.05, 18.67] | 0.231 |
| | <-3SD | 3 (18.8) | 2 (2.7) | 9.31 [1.18, 80.68] | **0.03**$^*$ |
| Baseline CD4 count | 0–200 | 2 (25) | 10 (18.5) | 1 | |
| | 201–500 | 5 (62.5) | 28 (51.9) | 0.89 [0.159, 5.35] | 1 |
| | 501–800 | 0 | 9 (16.7) | - | - |
| | ≥801 | 1 (12.5) | 7 (87.5) | 0.71 [0.05, 9.50] | 1 |
| Proteinuria | Yes | 7 (43.8) | 19 (26.8) | | |
| | No | 9 (56,2) | 52 (73.2) | 2.13 [0.70, 6.52] | 0.180 |
| *eGFR< 90ml/min/1.73m$^2$ | Yes | 8 (50) | 21 (28.8) | | |
| | No | 8 (50) | 52 (71.2) | 2.48 [0.82, 7.46] | 0.101 |
| *FEp>18% | Yes | 1 (6.2) | 1 (1.4) | | |
| | No | 15 (93.8) | 72 (98.6) | 4.80 [0.28, 81.10] | 0.233 |

*FEp = Fractional excretion of phosphate, *eGFR = estimated glomerular filtration rate, *BMI = Body mass Index.

assessing similar parameters for proximal tubular renal function used in our study, the prevalence was higher though. However the study population involved older patients compared to this study [12, 32]. Simultaneous measurement of urine albumin and protein may help differentiate tubular from glomerular proteinuria. Urine albumin/total protein ratio (uAPR) < 0.4 identifies tubular pathology in proteinuric patients and this was not measured in our study [33]. Although increased fractional excretion of phosphate, hypophosphatemia and normoglyacemic glycosuria are established markers of proximal tubular dysfunction and are easy to screen for, they are less sensitive than tubular protein excretions (NGAL, B2M and RBP) and have been suggested to be the most appropriate alternatives when these are not available [6, 34, 35]. In addition better markers for proximal tubular function (Retinol-binding protein, β-2 microglobulin and N-acetyl-β-d-glucosaminidase) could not be used due non-availability of tests and cost constraints. Lack of these more sensitive tests might have affected the results and

**Table 5. Factors associated with hypophosphatemia in children $\geq$ 14 years of TDF regimen for at least 6 months.**

| Variable | Category | Hypophosphatemia | | OR(95%CI) | p-value |
|---|---|---|---|---|---|
| | | Yes | No | | |
| Sex | Female | 2(33.3) | 46(47.9) | | |
| | Male | 4(66.7) | 50(52.1) | 0.54(0.10–3.11) | 0.487 |
| Current ART Reg. | TDF/3TC/EFV | 6(100) | 68(70.8) | | |
| | TDF/3TC/NVP | 0 | 13(13.5) | | |
| | TDF/3TC/PI | 0 | 15(15.6) | | |
| ART Regimen | 1st Line | 6(100) | 84(87.5) | | |
| | 2nd Line | 0 | 12(12.5) | | |
| Duration of Current Reg. | 0–12 | 1(16.7) | 5(5.2) | | |
| | 13–60 | 5(83.3) | 85(88.5) | | |
| | 61 plus | 0 | 6(6.3) | | |
| W.H.O Staging | Stage 1 | 0 | 19(29.7) | | |
| | Stage 2 | 1(25) | 14(21.9) | | |
| | Stage 3 | 3(75) | 24(37.5) | | |
| | Stage 4 | 0 | 7(10.9) | | |
| *BMI | Normal | 4(66.7) | 78(81.3) | | |
| | <-2SD | 0 | 5(5.2) | | |
| | <-3SD | 1(16.7) | 5(5.2) | | |
| | >1SD | 1(16.7) | 7(7.3) | | |
| | >2SD | 0 | 1(1.0) | | |
| Stunting | >-2SD | 5(83.3) | 66(68.8) | | |
| | <-2SD | 0 | 18(18.8) | | |
| | <-3SD | 1(16.7) | 12(12.5) | | |
| Baseline CD4 Count | 0–200 | 2(50) | 10(18.2) | | |
| | 201–500 | 1(25) | 27(49.1) | | |
| | 501–800 | 1(25) | 15(27.3) | | |
| | 801 plus | 0 | 3(5.5) | | |
| Proteinuria | Yes | 3(50) | 32(34.4) | | |
| | No | 3(50) | 61(65.6) | 1.91(0.36–9.99) | 0.358 |
| *eGFR< 90ml/min/1.73m$^2$ | Yes | 3(50) | 36(37.9) | | |
| | No | 3(50) | 59(62.1) | 1.64(0.31–8.56) | 0.426 |

**\*eGFR = estimated glomerular filtration rate, \*BMI = Body mass Index**

patients with proximal tubular dysfunction missed. A future prospective study on beta-2 microglobulin, aminoaciduria, hypercalciuria and NGAL as markers of tubular function is therefore recommended.

The prevalence of fractional excretion of phosphate >18% was low in this study compared to a study by Chadwick et al, in which 7% of the participants had FEp>18%. However, their sample size was smaller and included adults [12]. In a Swiss Cohort study, also done in adults, FEp >20% occurred in 11.5% of study patients which is much higher than our study [36]. Fractional excretion of phosphate varies with age with lower levels in children. In our study the FEp > 18% was used to define pathological fractional excretion of phosphate. It is possible that the age difference could explain the different findings [37, 38].

Hypophosphatemia reported in 8% of the participants was at least Division of Acquired Immunodeficiency Syndrome (DAIDS) toxicity grade 2. The prevalence of hypophosphatemia was higher than reported in other studies [39]. The long term consequence of this

| Variable | Category | Proteinuria | | Univariate | | Multivariate^ | |
| --- | --- | --- | --- | --- | --- | --- | --- |
| | | **Yes** | **No** | OR(95%CI) | p-value | **OR(95% CI)** | **p-value** |
| Age (years) | 0 -14 | 28 (43.1) | 63 (49.2) | 1 | | 1 | |
| | ≥15 | 37 (56.7) | 65 (50.8) | 1.28 [0.70; 2.34] | 0.420 | 1.25[0.65; 2.39] | 0.505 |
| | | | | | | | |
| Gender | Female | 27 (41.5) | 60 (46.9) | 1 | | 1 | |
| | Male | 38 (58.5) | 68 (53.1) | 1.24[0.68; 2.27] | 0.482 | 0.96[0.50; 1.84] | 0.899 |
| | | | | | | | |
| Current ART regimen. | TDF/3TC/EFV | 39 (60) | 96 (75) | 1 | | | |
| | TDF/3TC/NVP | 7 (10.8) | 19 (14.8) | 0.91[0.35, 2.33] | 0.839 | 0.91[0.35, 2.36] | 0.846 |
| | TDF/3TC/PI | 19 (29.2) | 13 (10.2) | 3.60[1.62, 7.99] | **0.002*** | **3.75[1.59; 86]*** | **0.003*** |
| | | | | | | | |
| Duration on current regimen in months. | 6 -12 | 7 (10.8) | 12 (9.4) | 1 | | 1 | |
| | 13 -60 | 56 (86.1) | 111 (86.7) | 0.86 [0.32; 2.32] | 0.773 | 1.25[0.42; 3.72] | 0.692 |
| | ≥61 | 2 (3.1) | 5 (3.9) | 0.69[0.10; 4.52] | 0.695 | 1.22[0.16; 9.06] | 0.848 |
| | | | | | | | |
| Stunting | >-2SD | 48 (73.8) | 104 (81.3) | 1 | | 1 | |
| | <-2SD | 11 (16.9) | 13 (10.2) | 1.83 [0.77; 4.39] | 0.173 | 1.52[0.60; 3.89] | 0.378 |
| | <-3SD | 6 (9.2) | 11 (8.6) | 1.18 [0.41; 3.38] | 0.756 | 1.01[0.33; 3.09] | 0.987 |

**Fig 1. Factors associated with proteinuria in HIV infected children, on TDF for at least 6 months, < 18 years old, (n = 193)**[†]. Children less than 18 years who weighed at least 25 kg and were on a TDF based regimen for at least six months were eligible for the study. Participants with pre-existing diabetes or hypertension were excluded from the study. A urine deep stick proteinuria followed by confirmatory urine protein: creatinine ratio to confirm proteinuria was performed for the participants. [†]6 samples not be processed due to a technical fault at the laboratory. *Statistically significant at α = 0.05. ^Only current ART regimen and stunting had p-value less than 0.25 in the univariate analysis. However, all variables were included in the multivariate analysis to control for any confounding effects from these variables.

hypophosphatemia in patients in the current study was unknown. In one study in children on TDF the phosphorous level normalised in four patients who had DAIDS toxicity grade 1 or 2 without discontinuation of therapy [39]. In a follow up study of 26 children for 132 months, hypophosphatemia occurred 72 months after commencing TDF but was of no clinical significance [40]. In a case series in adult patients on TDF, hypophosphatemia and osteomalacia necessitated discontinuation of therapy [41].

Patients in this study might still be at risk of rickets and or osteomalacia as they grow and continue taking TDF, hence we recommend continued clinical monitoring and follow up of patients with hypophosphatemia. Participants with stunting were more likely to have hypophosphatemia. However the numbers were small to make a general conclusion. The study cannot conclude on the contribution of malnutrition to hypophosphatemia rather than TDF. Future studies following up patients with hypophosphatemia who are stunted and on TDF may answer this question. A longitudinal prospective study of progressive impairment of renal function normalized by nutrition and HIV and phosphate excretion estimated on (a) timed (not 'spot' specimens) and (b) best expressed as tubular maximum of phosphate factored by GFR (i.e. TmP/GFR) is recommended for future studies. Normoglycaemic glycosuria was reported in only one child. This is much lower than reported in other studies in adults [12, 32]. Genetic polymorphism in the ATP binding cassette subfamily C2 (ABCC2) a gene that codes for organic anion transporters in the kidney has been reported as a risk factor for TDF

| Variable | Category | eGFR<90ml/min/1.73m² | | Univariate | | Multivariate^ | |
|---|---|---|---|---|---|---|---|
| | | Yes | No | OR (95%CI) | p-value | OR(95% CI) | p-value |
| Age (years) | 0 -14 | 29 (42.6) | 60 (49.2) | 0.77 [0.42, 1.40] | 0.387 | 0.78[0.33; 1.84] | 0.574 |
| | ≥ 15 | 39 (57.4) | 62 (50.8) | 1 | | 1 | |
| | | | | | | | |
| Gender | Female | 23 (33.8) | 64 (52.5) | 1 | | 1 | |
| | Male | 45 (66.2) | 58 (47.5) | 2.16 [1.17, 3.99] | **0.014*** | 1.32[0.57; 3.08] | 0.517 |
| | | | | | | | |
| Current ART Regimen. | TDF/3TC/EFV | 38 (55.9) | 99 (81.1) | 1 | | 1 | |
| | TDF/3TC/NVP | 9 (13.2) | 14 (11.5) | 1.67 [0.67, 4.19] | 0.270 | 1.05[0.29; 3.82] | 0.937 |
| | TDF/3TC/PI | 21 (30.9) | 9 (7.4) | 6.08[2.56, 14.45] | **<0.001*** | **4.43[1.32; 4.89]*** | **0.016*** |
| | | | | | | | |
| | | | | | | | |
| Stunting | >-2SD | 47 (69.1) | 101 (82.8) | 1 | | 1 | |
| | <-2SD | 11 (16.2) | 13 (10.7) | 1.82 [0.76, 4.36] | 0.180 | 1.56[0.41; 6.00] | 0.518 |
| | <-3SD | 10 (14.7) | 8 (6.6) | 2.69 [1.00, 7.24] | **0.051** | 2.17[0.49; 9.50] | 0.305 |
| | | | | | | | |
| Baseline CD4 Count | 201 − 500 | 18 (40) | 42 (56.8) | 1 | | 1 | |
| | 501 − 800 | 8 (17.8) | 16 (21.6) | 1.19 [0.43, 3.28] | 0.731 | 1.16[0.40; 3.36] | 0.788 |
| | ≤ 200 | 11 (24.4) | 13 (17.6) | 2.02 [0.76, 5.35] | 0.156 | 1.71[0.60; 4.87] | 0.313 |
| | ≥800 | 8 (17.8) | 3 (4) | 6.37[1.51, 6.79] | **0.012*** | **5.38[1.16; 5.04]*** | **0.032*** |

**Fig 2. Factors associated with reduction in eGFR in HIV infected children, on TDF for at least 6 months, < 18 years old, (n.190)[†].** Children less than 18 years who weighed at least 25 kg and were on a TDF based regimen for at least six months were eligible for the study. Participants with pre-existing diabetes or hypertension were excluded from the study. Following serum creatinine determination estimated glomerular filtration rate (eGFR) was calculated using the Schwartz formula. [†]8 samples not processed due to a technical fault at laboratory. *Statistically significant at α = 0.05. ^Only age had a p-value greater than 0.25 in the univariate model but it was also included in the multivariate model to account for any confounding effects age may have on eGFR.

associated nephrotoxicity in previous studies [42–44]. This was not investigated in this study that may also contribute to a lower prevalence of tubular dysfunction.

## Proteinuria

Urinary dipstick was positive for protein in 15.7% of participants and this doubled on urine protein/creatinine ratio. Clinical Practice Guidelines for Chronic Kidney Disease: Evaluation, Classification, and Stratification (KDOQI) guidelines recommend spot urine dipstick and urine protein/creatinine ratio to confirm proteinuria in children [8, 28]. Although a 24 hour urine collection is considered the gold standard for proteinuria, urine dipstick proteinuria with confirmatory urine protein creatinine ratio is also acceptable [45]. Urine albumin/total protein ratio (uAPR) < 0.4 identifies tubular pathology in proteinuric patients and this was not measured in our study [46]. The prevalence of proteinuria was much higher than what was reported in an earlier study in ART-naïve children in Zimbabwe, where persistent proteinuria was reported in 5% of the study population [10]. The difference in findings could be because the study by Dondo *et al* was on ART naïve younger patients (2–12 years). The prevalence is also lower than what was reported in studies elsewhere [12, 47]. Longer duration on TDF was a negative predictor of detection of proteinuria. In this study, the odds of having proteinuria in patients on TDF for >60 months was 0.69 CI (0.10; 4.52) compared to those with shorter duration. This was different to findings by Purswani *et al*, where TDF duration greater than three years was the single predictor of proteinuria [48]. Exposure to PI regimen was a positive predictor for having proteinuria. The findings are similar to other studies which showed the combination of a PI and TDF was associated with proteinuria [49, 50]. Since no other causes of proteinuria were assessed the authors do not conclude that PI is the cause of the proteinuria.

Gender and age were not predictors of proteinuria. Patients with advanced disease on this study were less likely to get proteinuria. This was different from findings by Dondo *et al.*, where WHO stage 3 or 4 disease was associated with proteinuria [10]. This could be due to the difference in study population. This study used the WHO clinical stage recorded at HAART initiation.

## Glomerular filtration rate

The prevalence of decreased eGFR was, comparable to an earlier study in ART naïve children in Zimbabwe [10]. However, studies in adult African patients had variable prevalence, 7.5% -86.5% [51–53]. The differences may be due to the different study methods, populations, adults compared to children and definition and cutoff of reduction in eGFR $< 60$ml/min/1.73m$^2$ versus $< 90$ml/min/1.73m$^2$. Combination with a PI was significantly associated with reduction in eGFR. On multivariate analysis use of PI was the only independent factor associated with reduction in eGFR. These findings were similar to other studies in children elsewhere [36, 54]. Advanced HIV disease, at least WHO stage 3 diseases was also associated with reduction in eGFR. Similar findings were reported by Dondo *et al.*, though patients were younger and ART naïve [10]. Severe stunting was significantly associated with reduction in eGFR. TDF may need to be used with caution in these children. Children on TDF for more than 60 months were 3.5 times likely to have reduction in eGFR. Cianflone *et al.*, reported an increase in reduction in GFR with longer duration of TDF in adult patients initiating HAART [55]. The World health organisation has now included tenofovir alafenamide (TAF) for children greater than 6 years weighing at least 25kg. The use of TAF versus TDF and effect of glomerular function will pave way for future studies[56].

## Conclusions

Proximal tubular dysfunction was uncommon among HIV infected children on TDF based regimen attending the outpatient HIV clinics at two tertiary hospitals in Harare. Hypophosphatemia was common and prevalence of proteinuria was high. Severe stunting was associated with hypophosphatemia. Combination with protease inhibitor was a risk factor for proteinuria. Reduction in eGFR was reported in children on a PI and those severely stunted. Combination with a PI drug regimen was the only independent factor associated with reduction in eGFR on multivariate analysis. Tenofovir alafenamide (TAF) that has different metabolism to TDF and is less nephrotoxic may be replacing the latter in the future [57] TAF's unique pharmacokinetic profile enables provision of lower required doses for antiviral efficacy. Lower concentrations reach renal tubules minimizing intracellular accumulation and mitochondrial damage hence less nephrotoxicity compared to TDF [58]. With future guidelines moving towards the use of integrase inhibitors the interaction with TDF/TAF and renal function is an area of future research. Children on TDF for longer duration for $> 60$ months may benefit from glomerular filtration measurement to assess their renal function. Further studies in children with hypophosphatemia and malnutrition may be of use in future. Patients with concomitant use of TDF and a PI regimen may benefit from targeted monitoring of glomerular function in resource-limited settings.

## Study limitations

This study was a cross-sectional study hence causality could not be determined. This study only looked at patients who were on TDF based regimen without comparing with patients who were on non–TDF. Future studies comparing proximal renal tubular function in children on both TDF and non-TDF based regimen are recommended to ascertain the causality of

proximal renal tubular dysfunction by TDF as this was not studied in the current study. Viral load was not done due to financial constraints but this could have helped to relate tubular and glomerular dysfunction to viral load.

## Supporting information

**S1 File. Original tables for Figs 1 and 2.**
(DOCX)

**S2 File. Laboratory protocol can be found at protocols.io** https://protocols.io/view/pone-d-19-35784-proximal-renal-tubular-function-in-bfzxjp7n **and DOI:** dx.doi.org/10.17504/protocols.io.bfzxjp7n.
(DOCX)

**S3 File. MDB file for the data sets** https://datadryad.org/stash/share/pRmK7SDWzk5Jnd3-usw2hKCGVe71OksxVKbxfTZv5PE **and DOI (doi:**10.5061/dryad.2fqz612mf**).**
(ACCDB)

**S4 File. Definition of terms.**
(DOCX)

## Acknowledgments

The authors would like to thank the participants and their caregivers for participating in the study. We would like to thank all the nurses who helped us with the work in the clinics. Our great felt gratitude goes to the laboratory staff for processing the samples and the statistician for helping with data analysis. We would also like to thank the department of Pediatrics for supervisory support and proofreading of findings and recommendations during the study period.

## Author Contributions

**Conceptualization:** Runyararo Mashingaidze-Mano, Mutsawashe F. Bwakura-Dangarembizi, Charles C. Maponga, Gene D. Morse, Tsitsi G. Monera-Penduka, Takudzwa J. Mtisi, Tinashe Mudzviti, Hilda A. Mujuru.

**Data curation:** Runyararo Mashingaidze-Mano.

**Funding acquisition:** Gene D. Morse.

**Investigation:** Runyararo Mashingaidze-Mano.

**Methodology:** Runyararo Mashingaidze-Mano, Mutsawashe F. Bwakura-Dangarembizi, Charles C. Maponga, Gene D. Morse, Tsitsi G. Monera-Penduka, Takudzwa J. Mtisi, Tinashe Mudzviti, Hilda A. Mujuru.

**Project administration:** Runyararo Mashingaidze-Mano, Mutsawashe F. Bwakura-Dangarembizi.

**Supervision:** Runyararo Mashingaidze-Mano, Mutsawashe F. Bwakura-Dangarembizi, Charles C. Maponga, Gene D. Morse, Hilda A. Mujuru.

**Writing – original draft:** Runyararo Mashingaidze-Mano.

**Writing – review & editing:** Runyararo Mashingaidze-Mano, Mutsawashe F. Bwakura-Dangarembizi, Charles C. Maponga, Gene D. Morse, Tsitsi G. Monera-Penduka, Takudzwa J. Mtisi, Tinashe Mudzviti, Hilda A. Mujuru.

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
