## [Decision Letter · Decision Letter 0]

14 Apr 2020

PONE-D-19-35784

Proximal renal tubular function in HIV-infected children on tenofovir disoproxil fumarate for treatment of HIV infection at two tertiary hospitals in Harare, Zimbabwe

PLOS ONE

Dear Dr. Rubyararo,

Thank you for submitting your manuscript to PLOS ONE. After careful consideration, we feel that it has merit but does not fully meet PLOS ONE’s publication criteria as it currently stands. Therefore, we invite you to submit a revised version of the manuscript that addresses the points raised during the review process.

We would appreciate receiving your revised manuscript by the 14. 05.2020. To enhance the reproducibility of your results, we recommend that if applicable you deposit your laboratory protocols in protocols.io, where a protocol can be assigned its own identifier (DOI) such that it can be cited independently in the future. For instructions see: http://journals.plos.org/plosone/s/submission-guidelines#loc-laboratory-protocols

We look forward to receiving your revised manuscript.

Kind regards,

Franziska Theilig, Prof.

Academic Editor

PLOS ONE

Journal Requirements:

2. Please include your tables as part of your main manuscript and remove the individual files. Please note that supplementary tables (should remain/ be uploaded) as separate "supporting information" files

Reviewers' comments:

Reviewer's Responses to Questions

**Comments to the Author**

1. Is the manuscript technically sound, and do the data support the conclusions?

Reviewer #1: Yes

Reviewer #2: Yes

2. Has the statistical analysis been performed appropriately and rigorously? 

Reviewer #1: Yes

Reviewer #2: Yes

3. Have the authors made all data underlying the findings in their manuscript fully available?

Reviewer #1: Yes

Reviewer #2: No

4. Is the manuscript presented in an intelligible fashion and written in standard English?

Reviewer #1: Yes

Reviewer #2: Yes

5. Review Comments to the Author

Reviewer #1: Even though the sample size is not so big, the study is well presented and gives relevant data in the use of TDF. In the methods they recruited children that were on that régimen for at least six months, but it would have been interesting to check the differences depending on how long they were on the treatment.

The conclusions are quite well presented and they also describe properly the most important limitations that it is that they did not compare a groups with other regimens.

In conclusion, the article is technically sound and gives the data needed for the conclusions and the author’s know and stand clear the limitations of the study.

Reviewer #2: Dr Mashingaidze-Mano and colleagues studied 198 children on TDF-based ARV therapy. This is a fantastic report and important contribution to the literature. The finding that 1/3 had abnormal proteinuria is extremely important and appears to be downplayed somewhat in the authors conclusions. The authors conclude a “low prevalence of proximal tubular dysfunction”. Proteinuria (particularly non-albumin proteinuria that typifies TDF toxicity) is a marker of tubular dysfunction.

Most importantly: I believe proteinuria (when confirmed by UPC ratio) should be included as a tubular abnormality and the entire prevalence of “tubular dysfunction” recalculated.

The intro is too long and the information about genetic deteminants (ABCC2) should be moved to discussion

Severe stunting should be defined

Define proteinuria cutoffs used for UPC ratios

Spell out DAIDS on first use

This article is generally well written but the discussion is confusing when discussing predictors of proteinuria. This should be reworded: “Longer duration on TDF was not a good predictor of detection of proteinuria. In this study, the odds of having proteinuria in patients on TDF for >60 months was 0.69 CI (0.10; 4.52) compared to those with shorter duration.” Do you mean “negative predictor” since the OR was < 1?? The following sentence is also confusing: “Exposure to PI regimen was a good predictor for having proteinuria.” This is not conventional language for discussing predictors.

TAF should be spelled out in first use and authors should explain differences from TAF to TDF, explain to reader why less nephrotoxic

6. PLOS authors have the option to publish the peer review history of their article (what does this mean?). If published, this will include your full peer review and any attached files.

Reviewer #1: No

Reviewer #2: No

---

## [Author Response · Author response to Decision Letter 0]

26 May 2020

The reviewers comments have been responded to. A rebuttal letter has been included to that effect.

---

## [Decision Letter · Decision Letter 1]

23 Jun 2020

Proximal renal tubular function in HIV-infected children on tenofovir disoproxil fumarate for treatment of HIV infection at two tertiary hospitals in Harare, Zimbabwe

PONE-D-19-35784R1

Dear Dr. Runyararo,

We’re pleased to inform you that your manuscript has been judged scientifically suitable for publication and will be formally accepted for publication once it meets all outstanding technical requirements.

Kind regards,

Franziska Theilig, Prof.

Academic Editor

PLOS ONE

Additional Editor Comments (optional):

Reviewers' comments:

Reviewer's Responses to Questions

**Comments to the Author**

1. If the authors have adequately addressed your comments raised in a previous round of review and you feel that this manuscript is now acceptable for publication, you may indicate that here to bypass the “Comments to the Author” section, enter your conflict of interest statement in the “Confidential to Editor” section, and submit your "Accept" recommendation.

Reviewer #2: All comments have been addressed

2. Is the manuscript technically sound, and do the data support the conclusions?

Reviewer #2: Yes

3. Has the statistical analysis been performed appropriately and rigorously? 

Reviewer #2: Yes

4. Have the authors made all data underlying the findings in their manuscript fully available?

Reviewer #2: Yes

5. Is the manuscript presented in an intelligible fashion and written in standard English?

Reviewer #2: Yes

6. Review Comments to the Author

Reviewer #2: (No Response)

7. PLOS authors have the option to publish the peer review history of their article (what does this mean?). If published, this will include your full peer review and any attached files.

Reviewer #2: No

---

## [Editor Report · Acceptance letter]

25 Jun 2020

PONE-D-19-35784R1 

Proximal renal tubular function in HIV-infected children on tenofovir disoproxil fumarate for treatment of HIV infection at two tertiary hospitals in Harare, Zimbabwe 

Dear Dr. Mashingaidze-Mano:

I'm pleased to inform you that your manuscript has been deemed suitable for publication in PLOS ONE. Congratulations! Your manuscript is now with our production department. 

Kind regards, 

on behalf of

Dr. Franziska Theilig 

Academic Editor

PLOS ONE